# Efficacy and safety of acupuncture treatment for post-stroke depression: A protocol for systematic review and meta-analysis

**Demin Kong**[1‡]**, Yangyang Li**[1‡]**, Wei Zou**[2]*

**1** Heilongjiang University of Chinese Medicine, Harbin, China, **2** First Affiliated Hospital Heilongjiang University of Chinese Medicine, Harbin, China

‡ DK and YL are contributed equally to this work and are co-first authors.
* kuangzou1965@163.com

## Abstract

### Background

Post-stroke depression is a common complication of stroke, with a high incidence rate and low recognition rate. Many patients do not receive effective intervention at the onset, which affects subsequent treatment outcomes. Post-stroke depression not only impacts the patient's mental well-being but also increases the risk of stroke recurrence and poor prognosis. Therefore, it has become a significant public health concern. Acupuncture has gained significant popularity in the treatment of post-stroke depression. However, there are inconsistent clinical research results regarding its efficacy and safety. This systematic review aims to gather and critically assess all available evidence regarding the effectiveness and safety of acupuncture in the treatment of post-stroke depression in patients.

### Methods

We will conduct thorough searches for relevant studies in multiple electronic databases (PubMed, Embase, Cochrane Library, China National Knowledge Infrastructure, VIP Database, Wan-fang Data and China Biomedical Database). Our search scope will encompass studies published from the inception of each database until September 2023. To evaluate the potential bias in all the included studies, we will adhere to the guidelines offered in the Cochrane Handbook. The total effective rate will be the primary outcome. To conduct a systematic review, we will employ RevMan 5.4 software.

### Results

This study will obtain efficacy and safety of acupuncture for the treatment of post-stroke depression.

### Conclusions

The conclusions of this study will provide evidence-based perspectives that can guide clinical decision-making regarding the practicality and recommended timing of using

**Data Availability Statement:** No datasets were generated or analysed during the current study. All relevant data from this study will be made available upon study completion.

**Funding:** The author(s) received no specific funding for this work.

**Competing interests:** The authors have declared that no competing interests exist.

acupuncture to treat post-stroke depression. Furthermore, this study will help advance the clinical application of acupuncture treatment for post-stroke depression and enhance its efficacy while ensuring patient safety.

## Introduction

### Description of the condition

Post-stroke depression (PSD) is a post-stroke syndrome characterized by mood disorders, loss of interest, anorexia, insomnia, and slowness of thinking in addition to the symptoms of stroke, and is usually accompanied by corresponding physical symptoms, and in severe cases, it may lead to fantasies, misanthropy, and suicidality [1, 2]. More than one-third of stroke survivors have been reported to suffer from the disease, with a cumulative incidence of 39%-52% within 5 years of stroke attack, and approximately 57% of patients recovering from PSD within one year [3, 4]. The psychosocial vulnerability model suggests that depressive symptoms in stroke survivors are mainly attributed to decreased engagement in daily activities during the recovery process [5]. Previous studies have indicated that PSD may stem from psychological stress responses triggered by post-stroke cognitive dysfunction, physical disability, or social isolation [6]. These patients face the risks of poor recovery of neurological function, increased disability and mortality rates, slow recovery and decreased quality of life, which seriously affects their recovery and quality of life, and creates additional burdens on their families and society [7]. The treatment of PSD remains extremely challenging due to its complex pathogenesis. Therefore, targeted interventions for PSD should be implemented in order to improve the progression of the patient's condition.

### Description and function of intervention

Currently, oral antidepressants such as selective serotonin reuptake inhibitors (SSRIs), serotonin and norepinephrine reuptake inhibitors (SNRIs), and norepinephrine and dopamine reuptake inhibitors (NDRIs) are commonly used clinically for the treatment of PSD, and their efficacy and safety remain controversial, with consideration of the patient's underlying medical condition, compliance, and adverse reactions to antidepressant medications, and the effectiveness of the clinical treatments is often more difficult to assess. In contrast, acupuncture, known as a 'green therapy,' has been proven effective and safe in treating PSD [8].

Acupuncture (AC), which is a being a minimally invasive technique, has been widely used in China and worldwide to improve the clinical symptoms and progression of various diseases, such as trigeminal neuralgia [9], migraine [10], essential hypertension [11], endometriosis-related pain [12], and male infertility [13]. In recent years, numerous studies have demonstrated the definite curative effect of acupuncture on PSD [14, 15]. Acupuncture can exchange information between immune-neurological-endocrine-microbial metabolism through the brain-intestinal axis, further balancing the structure of intestinal flora, maintaining intestinal homeostasis, improving the dysfunction of the hypothalamus-pituitary-adrenal axis, and inhibiting inflammatory responses, thus improving the symptoms of PSD patients [16].

### Why it is important to do this review

Acupuncture utilizes the advantages of Traditional Chinese Medicine (TCM) syndrome differentiation and treatment, while also reducing the side effects of Western medicine. It can improve patients' negative mood, alleviate somatic symptoms, and promote neurological

function recovery. Numerous randomized controlled trials and meta-analyses have consistently demonstrated the specific benefits of acupuncture treatment for patients with PSD [17, 18]. Despite the increasing use of acupuncture in PSD, there is still an absence of rigorously designed systematic reviews and meta-analyses to assess its efficacy and potential harms [17]. Hence, our objective is to conduct a comprehensive systematic review and meta-analysis of the existing literature to investigate the role of acupuncture in treating PSD and its potential benefits or risks for patients. The findings of this study aim to address the ongoing controversy surrounding the benefits or harms of acupuncture in patients with PSD and provide conclusive evidence for clinical decision-making.

## Methods

### Study registration

The current research project has been duly recorded in the PROSPERO (registration number CRD42023469165) of the International Prospective Register of Systematic Reviews. The study will strictly abide by the guidelines outlined in the Preferred Reporting Items for Systematic Reviews and Meta-Analysis Protocols (PRISMA-P) statement [19]. The detailed PRISMA-P checklist can be found in the S1 Checklist.

### Inclusion and exclusion criteria

**Type of study.**   We will include all Clinical RCTs without any restrictions on country; however, the studies must be in either English or Chinese. We will exclude reviews, animal experiments, theory discussions, case reports, conference articles, and other non-RCT studies.

**Type of participants.**   This study will only include participants who have post-stroke depression. Participants of any gender, age, occupation, education level, or severity will be considered. Nevertheless, patients with a psychiatric disorder or a history of taking psychotropic medications before their stroke will be excluded.

**Type of intervention.**   In the experimental group, AC should be administered while in the control group, non-AC should be utilized. Drug treatment was deemed suitable across all the groups.

**Type of outcome measures.**   The total effective rate of AC to treat PSD will be the primary outcome. The total effective rate refers to the proportion of effective treatment in all subjects. The total effective rate = remove invalid subjects/all subjects. Additionally, the secondary outcomes will encompass various dimensions such as the cure rate, Hamilton depression rating scale, adverse events incidence. The cure rate = cure subjects/all subjects.

**Search strategy.**   In order to conduct a comprehensive check, we will systematically explore 7 different databases including PubMed, Embase, Cochrane Library, Chinese National Knowledge Infrastructure (CNKI), VIP Database, Wan-fang Data, and Chinese Biomedical Database (CBM). The search will cover the entire timeline of each electronic database, from their establishment date to September 2023. To ensure an effective search, a combination of medical subject words and free words will be utilized, and tailored retrieval patterns will be applied to each database. For detailed information on the search strategy, please refer to the Table 1.

### Data collection

**Study selection.**   The process of selecting studies, encompassing screening of literature, extraction and management of data, as well as examination, will be carried out by two researchers. They will independently assess the titles, abstracts, and keywords of all retrieved studies, identifying the trials that fulfill the inclusion criteria. For further evaluation, the

**Table 1. The search strategy for PubMed.**

| ORDER | STRATEGY |
|---|---|
| #1 | Search: "Post-stroke depression"[Mesh] |
| #2 | Search: "PSD"[Title/Abstract] OR "Post-stroke, Depression" [Title/Abstract] |
| #3 | #1 OR #2 |
| #4 | Search: "Acupuncture"[Title/Abstract] OR "AC, Acupuncture" [Title/Abstract] OR "Needle"[Title/Abstract] OR "Filiform needle" [Title/Abstract] |
| #5 | Search: "randomized controlled trial"[Publication Type] OR "RCT randomized controlled"[Publication Type] OR "random allocation" [Title/Abstract] OR "allocation, random"[Title/Abstract] OR "randomized, controlled"[Title/Abstract] OR "clinical trial" [Title/Abstract] |
| #6 | Search: "humans"[MeSH Terms] NOT "animals"[MeSH Terms] |
| #7 | #5 AND #6 |
| #8 | #3 AND #4 AND #7 |

PSD, post-stroke depression; AC, acupuncture; RCT, randomized controlled trial.

complete texts of all potentially relevant studies will be acquired. In case of any disagreements, the two researchers will engage in discussion with the involvement of the third author. The Fig 1 illustrates the workflow for the study selection process.

**Assessment of risk of bias in included studies.** The risk of bias in all the studies included will be assessed using the Cochrane Handbook. The evaluation and description of risks will be based on the following 6 items: generation of random sequences, allocation concealment, blinding, incomplete outcome data, selective reporting, and other biases. Risk levels will be categorized into three: "low risk of bias", "high risk of bias", and "unclear risk of bias".

**Dealing with missing data.** In the event of inadequate or absent trial data, the corresponding author will receive an email notification for further information. If the required data is still not obtainable or if the author cannot be reached, these studies will be excluded from our analysis. We will perform a restricted analysis utilizing the accessible data and deliberate on the possible ramifications of any missing information.

## Data synthesis

We will utilize RevMan 5.4 statistical software for data integration and Meta analysis. Funnel plots and EGGER tests will be created using the R language. Dichotomous variables will be expressed using relative risk (RR), while continuous variables will be expressed using mean difference (MD). If different measurement tools are used for the same variable, standardized mean difference (SMD) will be used for analysis. The interval estimate will be expressed as a 95% confidence interval (CI). The fixed effect model (FEM) will be used when the homogeneity between studies is good ($P \geq 0.1$, $I^2 \leq 50\%$). In cases where there is significant heterogeneity between studies, we will analyze the reasons and conduct subgroup analysis or sensitivity analysis by eliminating literature and using other methods. If the heterogeneity remains high ($P<0.1$, $I^2>50\%$), the random effects model (REM) will be selected. For studies that are not suitable for meta-analysis, qualitative analysis will be used. $P<0.05$ means the difference is statistically significant.

## Subgroup analysis

To determine heterogeneity, subgroup analyses are performed when a large number of subgroup studies exist. This analysis includes different acupuncture techniques, point selection, and consideration of factors such as disease duration and severity.

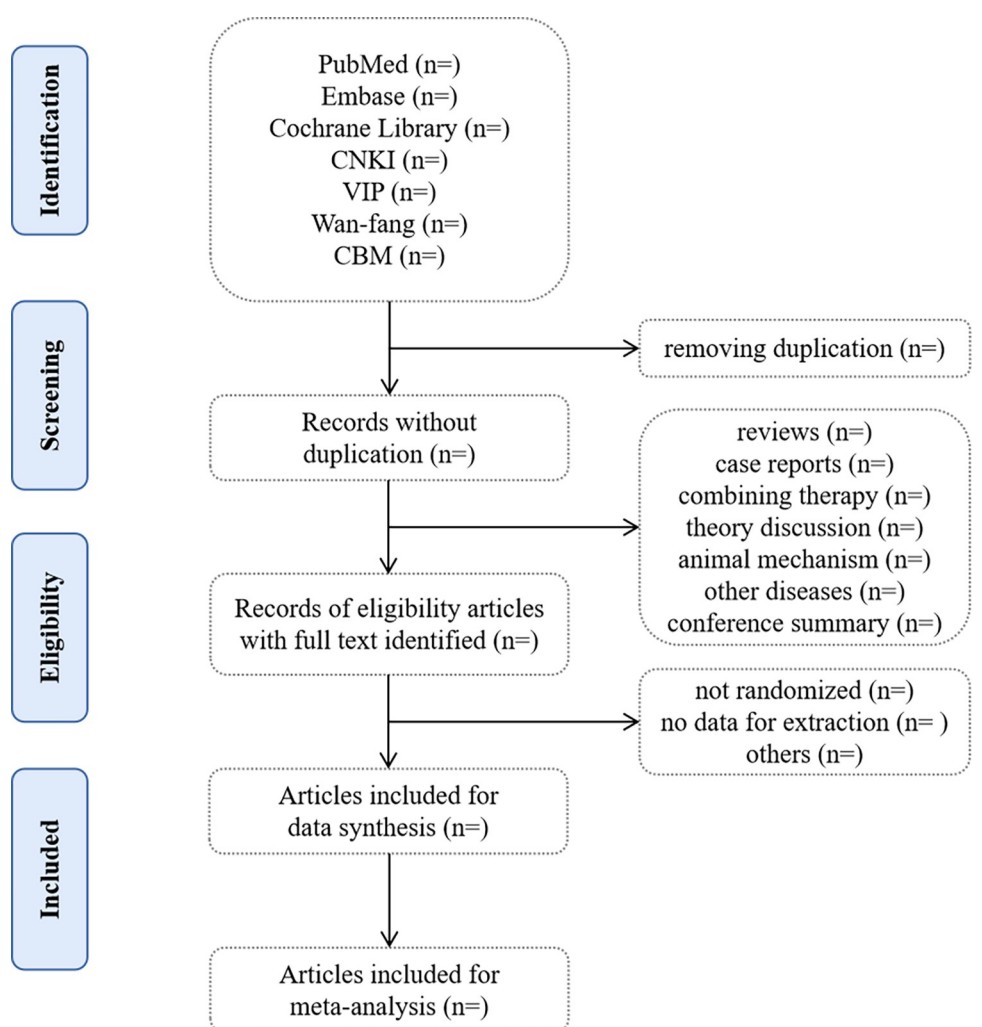

**Fig 1. The PRISMA flow diagram of the study selection process.**

## Sensitivity analysis

In order to validate the stability of the initial findings in the context of high heterogeneity, sensitivity analyses will be performed on the following different items: type of study, age difference, gender difference, study quality, treatment duration, treatment frequency and quality of heterogeneity. Reassessment of other relevant factors such as potential risk of bias, methodological quality and missing data will also be included.

## Publication bias

In case the meta-analysis incorporates over 10 studies, we will assess the presence of publication bias using the EGGER regression test. The assessment outcomes will then be exhibited through the utilization of funnel plots.

## Grading the quality of evidence

The quality of evidence will be assessed based on the Grading of Recommendations, Assessment, Development, and Evaluation (GRADE) [20]. The evaluation scale encompasses four

levels, namely high, moderate, low, and very low. Classification criteria: high (no evidence of downgrade), moderate (1 evidence of downgrade), low (2 evidence of downgrade), very low (more than 2 evidence of downgrade). The total clinical effective rate can be categorized into two levels, namely level 3 and level 4, based on the efficacy evaluation standards. Level 3 includes cure/markedly effective, effective, and ineffective outcomes, while level 4 includes cure, markedly effective, effective, and ineffective outcomes.

## Ethics and dissemination

Ethical approval is not necessary because our study is not linked to individual patient data. In addition, the study findings will be published by peer reviewed journals.

## Discussion

PSD, defined as post-stroke depressive condition, is the most common psychiatric problem following stroke, which reduces patients' quality of life and ability to recover [21]. TCM believes that PSD is an emotional disease that occurs on the basis of stroke due to a variety of reasons that lead to emotional disorders, which always belongs to the category of 'depression due to illness'. After a stroke, patients' motor function of the limbs, language and thinking ability and cognitive function will be damaged to varying degrees, which results in the gradual decline of the patients' ability to take care of themselves in their daily life and their ability to socialize, as well as the marginalization of society. As a result, the patient's self-care ability in daily life, social ability and social marginalization are gradually reduced, leading to symptoms such as being sentimental, having a bad mood, being depressed, and being lazy and less active. The American Heart Association recommends that antidepressants be administered to PSD patients and continued for at least 6 months after recovery [22]. However, studies have indicated that long-term use of SSRI antidepressants may increase the risk of stroke and myocardial infarction and is associated with increased all-cause mortality [23]. Therefore, there is an urgent need for a 'green treatment' to help patients reduce the adverse effects of antidepressants and improve neurological dysfunction, thereby alleviating medical stress.

Acupuncture as a uniquely holistic approach to treatment, with the ability to tailor the acupoint protocol to the patient's specific symptoms, may make acupuncture a uniquely supportive therapy for many stroke patients and may influence clinical decision-making regarding the use of acupuncture in post-stroke patients [15]. In recent years, acupuncture has been increasingly used in post-stroke care, with wrist-ankle acupuncture (WAA) improving the antidepressant effect of fluoxetine and reducing neurologic and gastrointestinal adverse drug reactions, and the combination of which is more effective in treating PSD [24]. Shao-Hua Zhang and colleagues have also demonstrated the effectiveness of acupuncture in treating PSD and alleviating patient symptoms [25]. However, due to insufficient evidence to support these claims, results of clinical studies on the effectiveness and safety of acupuncture in treating patients with PSD have been inconsistent. This leads to the question of whether acupuncture can be used to treat PSD, and when and how to make it more effective. Therefore, the aim of this study was to evaluate the efficacy and safety of acupuncture for the treatment of patients with PSD through systematic reviews and meta-analysis, in order to provide a basis for clinical evidence-based medical decision-making.

## Supporting information

**S1 Checklist. The PRISMA-P checklist.**
(DOCX)

## Acknowledgments

The authors would like to thank those who provided comments on the revision of this review.

## Author Contributions

**Conceptualization:** Demin Kong, Wei Zou.

**Data curation:** Demin Kong, Yangyang Li.

**Formal analysis:** Demin Kong, Yangyang Li.

**Investigation:** Demin Kong, Yangyang Li, Wei Zou.

**Methodology:** Demin Kong, Yangyang Li.

**Project administration:** Demin Kong, Wei Zou.

**Resources:** Demin Kong, Yangyang Li.

**Software:** Demin Kong, Yangyang Li.

**Supervision:** Wei Zou.

**Validation:** Demin Kong, Yangyang Li, Wei Zou.

**Visualization:** Demin Kong, Yangyang Li, Wei Zou.

**Writing – original draft:** Demin Kong.

**Writing – review & editing:** Demin Kong, Wei Zou.

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
