## [Decision Letter · Decision Letter 0]

3 Jan 2024

PONE-D-23-34488Efficacy and safety of acupuncture treatment for post-stroke depression:A protocol for systematic review and meta-analysisPLOS ONE

Dear Dr. Zou,

Thank you for submitting your manuscript to PLOS ONE. After careful consideration, we feel that it has merit but does not fully meet PLOS ONE’s publication criteria as it currently stands. Therefore, we invite you to submit a revised version of the manuscript that addresses the points raised during the review process.

Thank you for submitting the following manuscript to PLOS ONE.Please revise the manuscript according to the reviewers' comments and upload the revised file.==============================

We look forward to receiving your revised manuscript.

Kind regards,

Yung-Hsiang Chen, Ph.D.

Academic Editor

PLOS ONE

Journal Requirements:

2. Please clarify the search dates, as your manuscript states both 'until September 2023' and 'to September 20, 2024

5. We notice that your supplementary figures are uploaded with the file type 'Figure'. Please amend the file type to 'Supporting Information'. Please ensure that each Supporting Information file has a legend listed in the manuscript after the references list.

Additional Editor Comments:

Thank you for submitting the following manuscript to PLOS ONE.

Please revise the manuscript according to the reviewers' comments and upload the revised file.

Reviewers' comments:

Reviewer's Responses to Questions

**Comments to the Author**

1. Does the manuscript provide a valid rationale for the proposed study, with clearly identified and justified research questions?

Reviewer #1: Partly

Reviewer #2: Partly

2. Is the protocol technically sound and planned in a manner that will lead to a meaningful outcome and allow testing the stated hypotheses?

Reviewer #1: No

Reviewer #2: Yes

3. Is the methodology feasible and described in sufficient detail to allow the work to be replicable?

Reviewer #1: Yes

Reviewer #2: Yes

4. Have the authors described where all data underlying the findings will be made available when the study is complete?

Reviewer #1: Yes

Reviewer #2: No

5. Is the manuscript presented in an intelligible fashion and written in standard English?

Reviewer #1: No

Reviewer #2: Yes

6. Review Comments to the Author

You may also provide optional suggestions and comments to authors that they might find helpful in planning their study.

Reviewer #1: 1. This study failed to state why it is important to do this review. Several meta-analyses/protocols of meta-analysis have been published in recent years. However, there is no effective comparison between the previous meta-analysis and this one in the background or discussion.

2. The author needs to improve the English expression and grammar of the full text.

3. The total effective rate of AC to treat PSD was selected as the primary outcome in this study, but the author failed to state the definition of effective.

4. The protocol of data synthesis and Sensitivity analysis needs to be more detailed.

Reviewer #2: The manuscript, presenting a research protocol for a meta-analysis on the effectiveness and safety of acupuncture therapy in treating post-stroke depression, appears to be of good quality. The topic chosen is of considerable clinical significance. Although the initial quality of the manuscript is commendable, there are areas where further refinement could enhance its scientific rigor and clarity. Detailed observations and suggestions are provided below to assist in these potential improvements.

1. Innovativeness

Your manuscript mentions previous similar meta-analyses and describes their limitations. This is a good starting point. To strengthen the innovativeness of your manuscript, it is suggested to more elaborately detail how this study makes innovative contributions on this basis, and the importance of these innovations in the current field of research.

2. Structure of the Abstract

The abstract is a crucial part of the paper, aiming to concisely and clearly highlight the main points of the research. Regarding the background section, it is advised to streamline the current expression, retaining only information directly relevant to the research, to enhance the clarity and efficiency of the abstract.

3. Tense Issues

In lines 173 to 175, you refer to the characteristics of this study as a protocol. Regarding the use of tense, it is recommended to review and adjust to ensure consistency and accuracy in the text when describing the research design, methods, and expected outcomes.

4. Choice of SMD

When choosing the Standardized Mean Difference (SMD) as the measure of effect size, please provide a more detailed explanation as to why the Mean Difference (MD) is not applicable in this study.

5. Introduction to GRADE

In lines 188 to 190, the introduction to the GRADE assessment method seems overly brief. To better help readers understand how you have applied this method and its role in your research, please provide a more detailed explanation, including its assessment criteria and how it is applied in this study.

7. PLOS authors have the option to publish the peer review history of their article (what does this mean?). If published, this will include your full peer review and any attached files.

Reviewer #1: No

Reviewer #2: No

---

## [Author Response · Author response to Decision Letter 0]

1 Feb 2024

Dear Editors and Reviewers:

Thank you for your letter and for the reviewers' comments concerning our manuscript entitled “Efficacy and safety of acupuncture treatment for post-stroke depression: A protocol for systematic review and meta-analysis” (PONE-D-23-34488). Those comments are all valuable and very helpful for revising and improving our paper. We have studied comments carefully and have made correction which we hope meet with approval. And All changes made in the revised manuscript have been marked in red text. Major corrections in the paper and responses to editorial and reviewer comments follow:

Journal Requirements:

1)Please ensure that your manuscript meets PLOS ONE's style requirements, including those for file naming. The PLOS ONE style templates can be found at

Reply: The paper has been modified according to the journal format requirements.

2)Please clarify the search dates, as your manuscript states both 'until September 2023' and 'to September 20, 2024'.

Reply: Thank you for pointing out the issue. After careful consideration of the entire text, we have removed the section 'to September 20, 2024' for greater clarity.

3)Please provide a complete Data Availability Statement in the submission form, ensuring you include all necessary access information or a reason for why you are unable to make your data freely accessible. If your research concerns only data provided within your submission, please write "All data are in the manuscript and/or supporting information files" as your Data Availability Statement.

Reply: Provided .

4)PLOS requires an ORCID iD for the corresponding author in Editorial Manager on papers submitted after December 6th, 2016. Please ensure that you have an ORCID iD and that it is validated in Editorial Manager. To do this, go to ‘Update my Information’ (in the upper left-hand corner of the main menu), and click on the Fetch/Validate link next to the ORCID field. This will take you to the ORCID site and allow you to create a new iD or authenticate a pre-existing iD in Editorial Manager. Please see the following video for instructions on linking an ORCID iD to your Editorial Manager account: https://www.youtube.com/watch?v=_xcclfuvtxQ

Reply: The ORCID iD of the corresponding author has been added to 'Update my Information'.

5)We notice that your supplementary figures are uploaded with the file type 'Figure'. Please amend the file type to 'Supporting Information'. Please ensure that each Supporting Information file has a legend listed in the manuscript after the references list.

Reply: Changes have been made.

Reviewer #1: 

1)This study failed to state why it is important to do this review. Several meta-analyses/protocols of meta-analysis have been published in recent years. However, there is no effective comparison between the previous meta-analysis and this one in the background or discussion.

Reply: Currently, oral antidepressants such as selective serotonin reuptake inhibitors (SSRIs), norepinephrine reuptake inhibitors (SNRIs), and norepinephrine and dopamine reuptake inhibitors (NDRIs) are commonly used in clinical practice. However, the effectiveness and safety of treating post-stroke depression (PSD) using these antidepressants are still controversial. It is important to consider the patient's underlying condition, compliance, and potential adverse reactions to antidepressant drugs. Evaluating the clinical treatment effect can be challenging. Long-term use of SSRI antidepressants has been associated with an increased risk of stroke, myocardial infarction, and all-cause mortality. In contrast, acupuncture, known as a 'green therapy,' has been proven effective and safe in treating PSD. Acupuncture utilizes the advantages of Traditional Chinese Medicine (TCM) syndrome differentiation and treatment, while also reducing the side effects of Western medicine. It can improve patients' negative mood, alleviate somatic symptoms, and promote neurological function recovery. Therefore, studying the effectiveness of acupuncture treatment for PSD is of great significance, and the significance of the study is explained in the revised manuscript.

2)The author needs to improve the English expression and grammar of the full text.

Reply: Thanks for your suggestion. We have tried our best to polish the language in the revised manuscript. These changes will not influence the content and framework of the paper.

3)The total effective rate of AC to treat PSD was selected as the primary outcome in this study, but the author failed to state the definition of effective.

Reply: As suggested by the reviewer, we have added a definition of total effective rate in the Type of outcome measures section and highlighted it in red text.

4)The protocol of data synthesis and Sensitivity analysis needs to be more detailed.

Reply: Thanks for your suggestion. Data synthesis and Sensitivity analysis have been described in more detail.

Reviewer #2: The manuscript, presenting a research protocol for a meta-analysis on the effectiveness and safety of acupuncture therapy in treating post-stroke depression, appears to be of good quality. The topic chosen is of considerable clinical significance. Although the initial quality of the manuscript is commendable, there are areas where further refinement could enhance its scientific rigor and clarity. Detailed observations and suggestions are provided below to assist in these potential improvements.

1) Innovativeness

Your manuscript mentions previous similar meta-analyses and describes their limitations. This is a good starting point. To strengthen the innovativeness of your manuscript, it is suggested to more elaborately detail how this study makes innovative contributions on this basis, and the importance of these innovations in the current field of research.

Reply: TCM believes that PSD is an emotional disease that occurs on the basis of stroke due to a variety of reasons that lead to emotional disorders, which always belongs to the category of "depression due to illness". After a stroke, patients' motor function of the limbs, language and thinking ability and cognitive function will be damaged to varying degrees, which results in the gradual decline of the patients' ability to take care of themselves in their daily life and their ability to socialize, as well as the marginalization of society. As a result, the patient's self-care ability in daily life, social ability and social marginalization are gradually reduced, leading to symptoms such as being sentimental, having a bad mood, being depressed, and being lazy and less active. Acupuncture utilizes the advantages of TCM syndrome differentiation and treatment, while also reducing the side effects of Western medicine. It can improve patients' negative mood, alleviate somatic symptoms, and promote neurological function recovery. Therefore, studying the effectiveness of acupuncture treatment for PSD is of great significance, and the significance of the study is explained in the revised manuscript.

2) Structure of the Abstract

The abstract is a crucial part of the paper, aiming to concisely and clearly highlight the main points of the research. Regarding the background section, it is advised to streamline the current expression, retaining only information directly relevant to the research, to enhance the clarity and efficiency of the abstract.

Reply: Thanks for your suggestion. Abstracts have been streamlined to better highlight the main points of the paper's research.

3) Tense Issues

In lines 173 to 175, you refer to the characteristics of this study as a protocol. Regarding the use of tense, it is recommended to review and adjust to ensure consistency and accuracy in the text when describing the research design, methods, and expected outcomes.

Reply: We feel sorry for our carelessness. In our resubmitted manuscript, the tense issues in the subgroup analysis section has been revised. Thanks for your correction.

4) Choice of SMD

When choosing the Standardized Mean Difference (SMD) as the measure of effect size, please provide a more detailed explanation as to why the Mean Difference (MD) is not applicable in this study.

Reply: After gaining an in-depth understanding of the definitions of SMD and MD, we believe that this study should use MD as a measure of effect size. At the same time, it is stated in the revised manuscript that if different measurement tools are used for the same variables, SMD will be used for analysis. Thank you for your correction, which allowed us to make timely adjustments to the statistical methods of the protocol.

5) Introduction to GRADE

In lines 188 to 190, the introduction to the GRADE assessment method seems overly brief. To better help readers understand how you have applied this method and its role in your research, please provide a more detailed explanation, including its assessment criteria and how it is applied in this study.

Reply: Thank you for your suggestion. We have provided a more detailed explanation of the GRADE evaluation method, evaluation criteria, and its application in this study in the revised manuscript.

We tried our best to improve the manuscript and made some changes in the manuscript.

We appreciate for Editors and Reviewers' warm work earnestly, and hope that the correction will meet with approval. Once again thank you very much for your comments and suggestions.

Yours sincerely,

Wei Zou

---

## [Decision Letter · Decision Letter 1]

5 Mar 2024

Efficacy and safety of acupuncture treatment for post-stroke depression:A protocol for systematic review and meta-analysis

PONE-D-23-34488R1

Dear Dr. Zou,

We’re pleased to inform you that your manuscript has been judged scientifically suitable for publication and will be formally accepted for publication once it meets all outstanding technical requirements.

Kind regards,

Yung-Hsiang Chen, Ph.D.

Academic Editor

PLOS ONE

Additional Editor Comments (optional):

Your manuscript has been evaluated by external reviewers and members of the editorial board.

Congratulations on the acceptance of your manuscript, and thank you for your interest in submitting your work to PLOS ONE.

Reviewers' comments:

Reviewer's Responses to Questions

**Comments to the Author**

1. Does the manuscript provide a valid rationale for the proposed study, with clearly identified and justified research questions?

Reviewer #3: Yes

2. Is the protocol technically sound and planned in a manner that will lead to a meaningful outcome and allow testing the stated hypotheses?

Reviewer #3: Yes

3. Is the methodology feasible and described in sufficient detail to allow the work to be replicable?

Reviewer #3: Yes

4. Have the authors described where all data underlying the findings will be made available when the study is complete?

Reviewer #3: Yes

5. Is the manuscript presented in an intelligible fashion and written in standard English?

Reviewer #3: Yes

6. Review Comments to the Author

You may also provide optional suggestions and comments to authors that they might find helpful in planning their study.

Reviewer #3: The paper discusses the high incidence and low recognition rate of post-stroke depression (PSD), emphasizing its impact on patients’ mental well-being and the increased risk of stroke recurrence and poor prognosis. It highlights the public health concern posed by PSD. It examines the popularity and inconsistent clinical research results regarding the efficacy and safety of acupuncture as a treatment for PSD. The systematic review aims to gather and critically assess all available evidence on the effectiveness and safety of acupuncture in treating patients with PSD. The authors address the journal’s requirements and reviewers’ comments, making corrections and providing clarifications to improve the manuscript. They emphasize the significance of studying acupuncture for PSD and detail their responses to the editorial and reviewer comments. This section serves as an introduction to the study protocol for a systematic review and meta-analysis on acupuncture treatment for PSD. It sets the stage for the detailed methodology and expected outcomes of the review. The manuscript has undergone a thorough review, and the revisions have been commendably executed. I advocate considering this manuscript for publication.

7. PLOS authors have the option to publish the peer review history of their article (what does this mean?). If published, this will include your full peer review and any attached files.

Reviewer #3: No

---

## [Editor Report · Acceptance letter]

24 Apr 2024

PONE-D-23-34488R1 

PLOS ONE

Dear Dr. Zou, 

I'm pleased to inform you that your manuscript has been deemed suitable for publication in PLOS ONE. Congratulations! Your manuscript is now being handed over to our production team.

Kind regards, 

on behalf of

Dr. Yung-Hsiang Chen 

Academic Editor

PLOS ONE